# Optofluidic crystallithography for directed growth of single-crystalline halide perovskites

Xue-Guang Chen[1,2], Linhan Lin [2] ✉, Guan-Yao Huang[3], Xiao-Mei Chen[2], Xiao-Ze Li[2], Yun-Ke Zhou[2], Yixuan Zou[2], Tairan Fu [3], Peng Li [2], Zhengcao Li [1] ✉ & Hong-Bo Sun [2,4] ✉

Crystallization is a fundamental phenomenon which describes how the atomic building blocks such as atoms and molecules are arranged into ordered or quasi-ordered structure and form solid-state materials. While numerous studies have focused on the nucleation behavior, the precise and spatiotemporal control of growth kinetics, which dictates the defect density, the micromorphology, as well as the properties of the grown materials, remains elusive so far. Herein, we propose an optical strategy, termed optofluidic crystallithography (OCL), to solve this fundamental problem. Taking halide perovskites as an example, we use a laser beam to manipulate the molecular motion in the native precursor environment and create inhomogeneous spatial distribution of the molecular species. Harnessing the coordinated effect of laser-controlled local supersaturation and interfacial energy, we precisely steer the ionic reaction at the growth interface and directly print arbitrary single crystals of halide perovskites of high surface quality, crystallinity, and uniformity at a high printing speed of $10^2$ μm s$^{-1}$. The OCL technique can be potentially extended to the fabrication of single-crystal structures beyond halide perovskites, once crystallization can be triggered under the laser-directed local supersaturation.

Crystallization kinetics is a determinant of the crystallinity and properties of crystalline materials[1-4]. It is generally described by two key processes, i.e., nucleation and crystal growth. The former determines the timing and spatial location where crystallization occurs, and thus the amount and the spatial distribution of crystals[5-9]. In contrast, the crystal growth process is more relevant to the properties of grown materials as it directs the arrangement of atomic building blocks such as atoms or molecules into ordered structure[10,11]. It dictates the defect generation, morphology, symmetry, and surface quality of crystalline

materials and, thus, their properties[12-16]. For example, defect control is a key procedure during the growth of semiconductors as the concentration of trap states in the materials dominates the carrier mobility, quantum efficiency, and other carriers-relevant properties for their applications in light-emitting diodes, solar cells, photodetectors, and so on[17-22].

Fundamentally speaking, the growth kinetics, including growth rate and growth pathway, relies on the atomic diffusion and interaction at the growth interface[11,23-25]. The key ingredients that influence

[1]Key Laboratory of Advanced Materials (MOE), School of Materials Science and Engineering, Tsinghua University, Haidian, Beijing 100084, China. [2]State Key Laboratory of Precision Measurement Technology and Instruments, Department of Precision Instrument, Tsinghua University, Haidian, Beijing 100084, China. [3]Key Laboratory for Thermal Science and Power Engineering of Ministry of Education, Beijing Key Laboratory of CO2 Utilization and Reduction Technology, Tsinghua University, Beijing 100084, China. [4]State Key Laboratory of Integrated Optoelectronics, College of Electronic Science and Engineering, Jilin University, 2699 Qianjin Street, Changchun 130012, China. ✉e-mail: linlh2019@mail.tsinghua.edu.cn; zcli@tsinghua.edu.cn; hbsun@tsinghua.edu.cn

the growth kinetics are supersaturation and surface energy, the former of which can be controlled by the solution concentration and composition, as well as the environmental temperature during crystallization[26–28]. For example, inverse-temperature crystallization, which improves the supersaturation by increasing temperature, has been widely used for the preparation of high-quality perovskite single crystals[29,30]. In addition, the regulation of surface energy difference by tuning the chemical potential leads to anisotropic crystal growth[31]. However, all these existing strategies are limited to the global kinetics control, i.e., the growth rate of all the crystals in the solution can be accelerated or suppressed simultaneously, while regioselective and real-time control of the growth kinetics at high accuracy remains challenging so far.

The key to addressing this fundamental problem is to locally tune the supersaturation and surface energy at the growth interface. Laser-controlled crystallization can potentially address this fundamental problem. The key is to locally tune the supersaturation using a laser beam. For instance, crystallization of halide perovskites can be triggered in a laser-generated optical potential at the liquid/gas interface and the grown crystals exhibit good crystallinity and photoluminescence (PL) properties[32,33]. The reduction of solubility in the laser-controlled temperature field can also be harnessed for crystallization control, with the fluorescence lifetime of the grown materials comparable to that of the single crystals grown in bulk solution[34]. Moreover, laser-triggered acid-catalyzed hydrolysis can also initialize the crystallization of halide perovksites[35]. Nevertheless, precise control of the crystallization pathway remains challenging to fabricate

perovskite micropatterns. In this work, we develop an optical technique termed optofluidic crystallithography (OCL) by harnessing the light-matter interaction at the molecular scale, i.e., using a laser beam to direct the molecular motion and redistribution and to create a local supersaturation environment. Both the growth velocity and growth pathway can be defined in real time by steering the laser spot. We experimentally demonstrate the laser-directed growth of single-crystal halide perovskites into arbitrary micropatterns. Our results reveal that OCL is a promising technique to control the wet-chemical reaction and growth kinetics to produce high-quality single-crystal halide perovskites for applications in various optoelectronic and photonic devices.

## Results and discussion
### Working principle
To illustrate the concept of OCL, we select methylammonium lead tribromide (MAPbBr$_3$) as an example. As shown in Fig. 1a, a layer of saturated MAPbBr$_3$ precursor solution of 15 μm in thickness, i.e., the mixture of equivalently stoichiometric methylammonium bromide (MABr) and lead dibromide (PbBr$_2$) dissolved in $N,N$-dimethylformamide (DMF), is placed inside a chamber on a glass substrate. When a laser beam (515-nm femtosecond laser beam of 0.45 mW in optical power and 1.3 μm in beam size) is focused onto a perovskite crystal in the precursor solution, the photon is absorbed by the crystals, and part of the energy is converted to heat through non-radiative decay. Heat diffusion is limited by the low thermal conductivity of MAPbBr$_3$ (0.51 W m$^{-1}$ K$^{-1}$) and DMF (0.1816 W m$^{-1}$ K$^{-1}$)[36,37]. Consequently, a

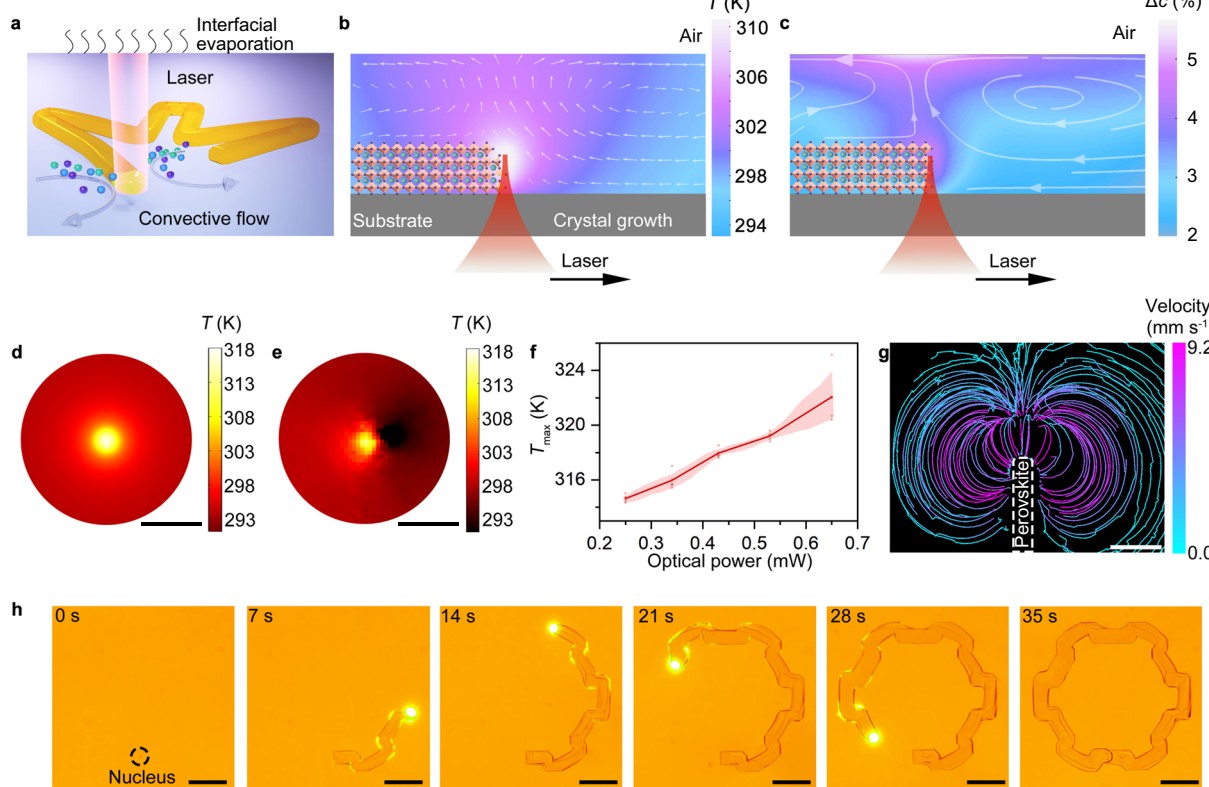

Fig. 1 | **Working principle of OCL. a** Schematic of OCL. The perovskite structure grows along the track of the laser spot. The interfacial evaporation and convection processes shown in the schematic create local supersaturation. **b** The simulation results of the temperature field and convective flow field. The white arrows represent the Marangoni convective flow. **c** Simulation results showing the degree of supersaturation (Δc) around the growth interface. The grey curves and arrows represent the simulated concentration flux. **d** The simulated in-plane temperature distribution when a MAPbBr$_3$ crystal is irradiated by a 515 nm femtosecond laser at

an optical power of 0.45 mW. **e** The mapping of in-plane temperature distribution recorded in the experiment. **f** The maximum temperature as a function of the incident laser power. The red line shows the average of the temperature. The red shade shows the standard deviation of the temperature. **g** The convective flow distribution was recorded by tracking the particle trajectories around the laser spot. **h** Successive optical images showing the laser direct printing of a single-crystal MAPbBr$_3$ structure. Scale bar: 50 μm. Source data are provided as a Source Data file.

thermal hotspot with a maximum temperature of 318 K is created around the perovskite crystal, with the maximum temperature tunable by the laser power (Fig. 1b, d–f). As the thickness of the precursor solution layer is thin, the local optical heating introduces a temperature gradient at the liquid/gas interface and drives Marangoni convection (see Fig. 1g) and interfacial evaporation. It is noted that the local evaporation reduces the amount of solvent molecules and improves the solute concentration, which is the key factor for the formation of supersaturation environment. The evaporation rate increased with the local temperature due to higher saturated vapor pressure. The Marangoni convection, together with the non-isothermal diffusion, redistributes the molecular species in the solution and creates a local supersaturation environment around the laser spot (Suppl. Note 1). Our simulation reveals that the coordinated effect significantly improves the local precursor concentration at the laser spot, with the supersaturation of precursors improved by ≈5% at a moderate temperature increment of 17 K (Fig. 1c). The local solute concentration can be further tuned by the laser power (Suppl. Fig. 1).

To experimentally demonstrate the OCL, we irradiated a focused laser spot onto a seed crystal, which was created by either laser-induced nucleation or spontaneous nucleation (see Methods), and steered the laser spot out of the crystal. At low optical power (0.1 mW), a mild improvement of local supersaturation accelerates the ionic reaction and leads to an improved spontaneous growth rate of the MAPbBr$_3$ crystal. In this situation, the average time for bonding atoms in the precursor solution to the crystal surface $t_b$ is significantly longer than the average time of the atomic rearrangement process $t_r$. The rearrangement process occurs efficiently to form a flat atomic layer before the formation of a second layer, leading to a layer-by-layer spontaneous growth mode. In the experiment, we observed the solid-liquid interface under laser irradiation migrating at 0.33 µm s$^{-1}$ to maintain the low surface energy (see Suppl. Movie 1). Above 0.15 mW, the local concentration is higher, and $t_b$ is significantly reduced ($t_b < t_r$). There are many randomly bound solute atoms bonded to the crystal surface as the atoms cannot relax efficiently to maintain the low-energy surface. Such a rapid liquid-to-solid phase transition leads to the exposure of atomically rough interfaces at the growth interface with much higher surface energy, which reduces the thermodynamic barrier of the liquid-to-solid phase transition and dramatically improves the growth rate (see detailed analysis in Suppl. Note 2). In our experiment, the growth rate reaches 0.1 mm s$^{-1}$ at 0.42 mW (see Suppl. Fig. 2). More importantly, different from the migration of the entire low-index surface observed at the low optical power, the growth pathway can be arbitrarily defined by the scanning trajectory of the laser spot. It is noted that the rapid ionic depletion around the growth interface is overcome by the strong Marangoni convective flow, which rebuilds the supersaturation environment by successive ionic feeding. As a demonstration, we built a two-dimensional micro-gear structure of 200 µm in diameter by a single laser scan in 35 s (see Fig. 1h and Suppl. Movie 2). The grown MAPbBr$_3$ structure is a single crystal without observation of any grain boundary or laser-induced damage (see Suppl. Note 3).

To verify the above mechanism, we carried out a couple of control experiments. Using a 660 nm continuous-wave laser which cannot be absorbed by the MAPbBr$_3$ crystal, we cannot observe the light-directed crystal growth and exclude optical force as the driving force to concentrate the precursors (see Suppl. Note 4). The repetition frequency of femtosecond laser also influences the quality of the grown structures. Normally, low repetition frequency is preferred in our experiments to suppress the defect generation (Suppl. Fig. 3). However, microbubbles can be generated at lower repetition frequency (e.g., 0.1 MHz) due to the higher pulse energy, and laser-directed printing can be interrupted. It is also known that the solubility of most perovskite precursors is a function of the environmental temperature. For MAPbBr$_3$ in the solvent of DMF, the solubility is reduced at an increased temperature, which may also lead to a supersaturation condition[26]. However, we found that OCL is also effective in directing the growth of MAPbBr$_3$ in the solvent of dimethyl sulfoxide (DMSO), whose solubility is increased with temperature, revealing that the temperature-dependent solubility is not a dominating factor (see Suppl. Note 5). It is noted that the thickness of the precursor solution is critical as it determines the local evaporation rate at the laser spot and the formation of local supersaturation (see Suppl. Note 6). The increase in precursor thickness reduces the temperature gradient at the liquid-gas interface and strongly suppresses the convective flow velocity. Thus, the local evaporation rate is reduced, and the precursor concentration cannot be significantly improved. In our experiments, the typical solution thickness is 8–15 µm. The laser-directed growth of halide perovskites cannot be observed when the solution thickness is above 30 µm.

## Laser-controlled crystallization kinetics

Since the growth pathway of single-crystal halide perovskites is defined by laser scanning trajectory during OCL, a number of high-index facets with high surface energy are exposed to the saturated precursor solution. As displayed in Fig. 2a, the grown MAPbBr$_3$ semi-circular structure has an arbitrary crystal orientation on the side surface. Such high-index facets are thermodynamically unstable, and rapid spontaneous growth on these surfaces is observed, which changes the morphology of the crystal until low-index facets dominate. As shown in Fig. 2b, the laser-printed semicircular micropattern expands rapidly in 2 min, which indicates that precise morphology control remains challenging. To overcome this technical difficulty, we introduce surface ligands (the volume ratio between oleylamine and oleic acid is 3:1, see Methods and Suppl. Note 7 for details) into the precursor solution to passivate the crystal surface. As shown in Fig. 2c, the surface ligands spontaneously adsorb onto the crystal surface after crystallization, which significantly reduces the surface energy. At the growth interface, the femtosecond laser triggers the desorption of surface ligands and subsequent rapid crystal growth with the local supersaturation[38]. After laser scanning, re-adsorption of surface ligands occurs to terminate the spontaneous and uncontrollable crystal growth. At a ligand concentration of 1:300 (the volume ratio between oleylamine and ligand-free precursor), we can see that the printed MAPbBr$_3$ semicircular structure well maintains the laser-defined morphology without any shape change after 40 min (see Fig. 2d and Suppl. Movie 3).

It is noted that the adsorption-desorption trade-off can be tuned by the concentration of surface ligands. At a high ligand concentration, the adsorption rate is improved, and the crystal growth rate is suppressed. The excessive amount of surface ligands in the precursor solution also provides nucleation sites and reduces the nucleation-free energy. When the nucleation rate exceeds the growth rate (e.g., at a ligand concentration above 1:80), laser-induced nucleation occurs, and a polycrystalline structure is obtained (see Fig. 2e, f and Suppl. Movie 3). The laser-controlled crystallization kinetics of halide perovskites at different ligand concentrations and optical powers is summarized in Fig. 2g. We can see that the laser direct printing of single-crystal (LDPSC) is achieved when the laser power exceeds the power threshold ($P_{th}$) and when the ligand desorption rate is above the adsorption rate. For precise morphology control, the ligand concentration should be improved to reduce the interfacial free energy until the spontaneous growth rate $v_{sp}$ becomes zero. However, at an excessive surface ligand concentration, the laser-induced high nucleation rate will lead to the mono-crystalline to polycrystalline transformation. We also measured the maximum growth rate during LDPSC as a function of the surface ligand concentration and the incident laser power (Fig. 2h). We can see that the single-crystal growth velocity is significantly decreased at an improved ligand concentration. From another perspective, the laser power has to be improved to maintain the crystal growth at the same velocity after the addition of surface ligands, which further verifies the passivation

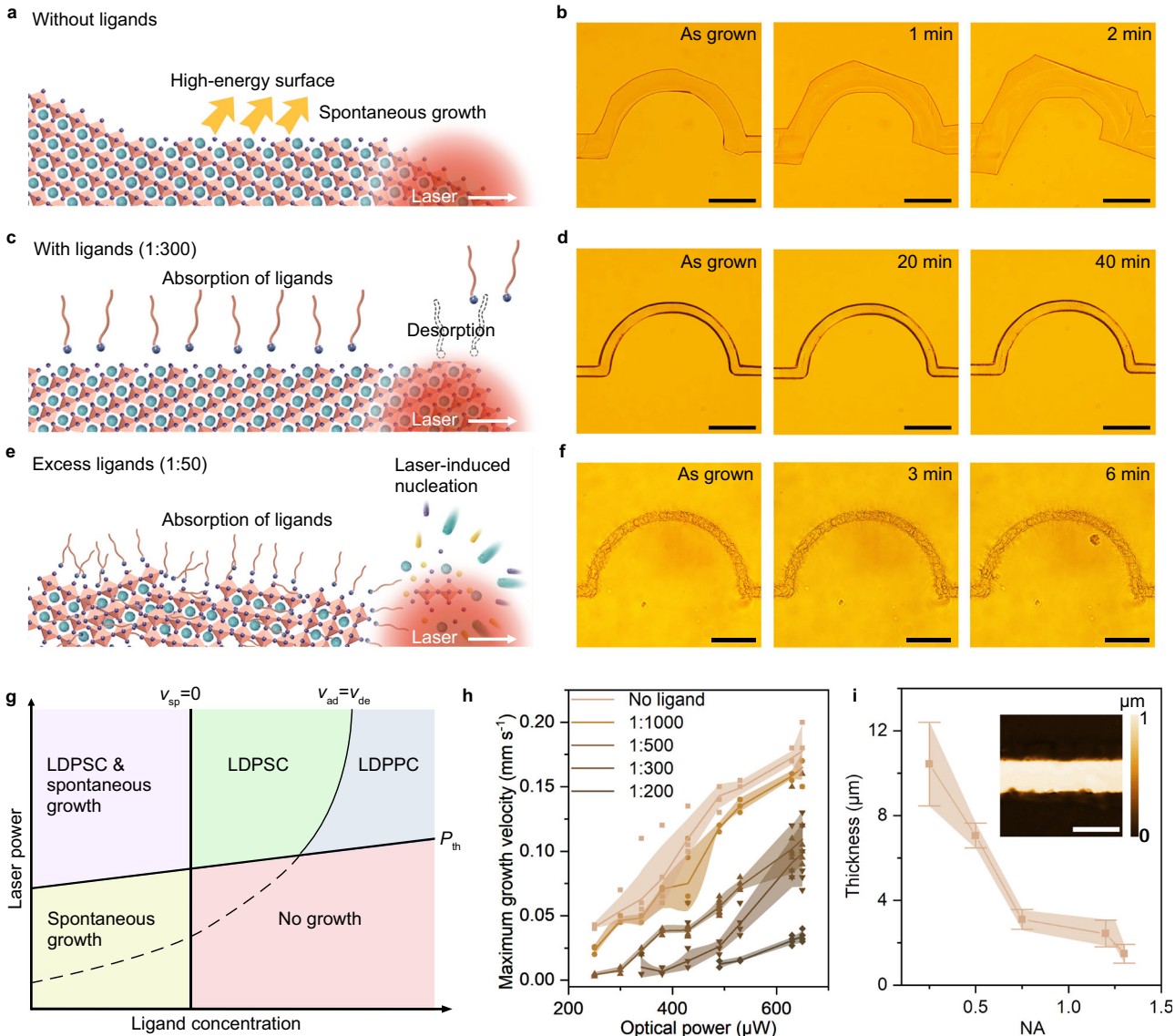

**Fig. 2 | Laser-controlled crystallization kinetics. a** Schematic showing the spontaneous growth on high-index facets after laser direct printing without surface ligands. **b** Experimental observation of rapid spontaneous growth on a semicircular pattern created by OCL. **c** Schematic showing the laser-controlled desorption-adsorption trade-off of surface ligands for precise morphology control. **d** Optical images showing the printed single-crystal semicircular structure passivated by surface ligands without spontaneous growth. **e** Schematic showing the laser-induced nucleation at a high concentration of surface ligands and the growth of polycrystalline structure. **f** Optical images showing the printed polycrystalline semicircular structure without spontaneous growth. Scale bar: 50 μm. **g** The phase diagram showing the laser-steered crystallization at different optical powers and ligand concentrations (LDPSC: laser direct printing of single crystal; LDPPC laser direct printing of polycrystal). $P_{th}$ is the optical power threshold of the incident laser, $v_{sp}$ is the spontaneous growth rate, $v_{ad}$ and $v_{de}$ represent the ligand adsorption and desorption rate, respectively. **h** The maximum growth velocity of LDPSC as a function of the surface ligand concentration and the optical power. The shades show the corresponding standard deviation. **i** The thickness of printed single-crystal halide perovskites using objectives of different NA. The inset panel shows the white light interferometry image of a single crystal printed with an objective of NA = 1.3. Scale bar: 5 μm. The error bars represent the standard deviations. $n$ = 12 independent replicates. Source data are provided as a Source Data file.

effect of the surface ligands. The growth rate of 0.1 mm s⁻¹ at the optimal ligand concentration (1:300) also reveals that OCL provides an efficient approach for the fabrication of single-crystal halide perovskites. The printing throughput can be easily scaled up through engineering of the laser beam, e.g., replacing the focus laser spot with a centimeter-scale linear optical landscape could dramatically improve the throughput and facilitate large-scale fabrication of single-crystal halide perovskite thin films.

The thickness of halide perovskites is important as it is influential on the carrier separation or recombination efficiency for various device applications such as solar cells and light-emitting diodes. In OCL, the thickness of the printed halide perovskite structures can be tuned by laser power, scan speed, and numerical aperture (NA) of the objective. Specifically, tuning the NA of objectives could shape the incident laser and thus the spatial distribution of the local temperature. As shown in Fig. 2i, the thickness of the printed halide perovskites decreases when the NA of the objectives is improved. A thickness of ≈1 μm is achieved at the high NA of 1.3 (also see the white light interferometry image in the inset). It can be further reduced by optimization of the printing parameters.

## Characterization of the halide perovskite single crystals

The precise optical manipulation of chemical reactions and crystallization kinetics through OCL provides a promising strategy for one-

step fabrication of arbitrary perovskite single crystals without any lithography process. As shown in Fig. 3a, we programed the scanning pathway of the focused laser spot and printed different MAPbBr$_3$ single crystals of complicated geometries with an overall crystal size of hundreds of micrometers. We can see that the morphology of the MAPbBr$_3$ crystals can be precisely defined by the laser trajectory without any crack observed, even when the scanning laser beam makes a sharp turn. The confocal fluorescence images show that the printed MAPbBr$_3$ structures exhibit excellent fluorescent properties with a uniform intensity distribution (Fig. 3b, c). For the micropatterns created in Fig. 3, the filling factor ranges from 26.6% to 32.3% for a single pattern. In principle, a filling factor of 100% can be achieved when the laser scanning pathway is designed to grow a continuous perovskite thin film. A more feasible strategy is to use a line-patterned laser beam to direct the growth kinetics. As shown in Suppl. Fig. 4, we used a concave cylindrical lens and a convex cylindrical lens to create a line pattern of 2 μm in width and 180 μm in length (after a 20× objective). Scanning of the line laser pattern can rapidly create large-area and dense perovskite thin film.

To investigate the surface quality of the printed MAPbBr$_3$ structure, we took a scanning electron microscopy (SEM) image of a laser-created arrow micropattern. From Fig. 3d–f, we can see that the crystal surface is clean and smooth, without observation of any grain boundary in the close-up images. The surface roughness measured by white light interferometry gives a value of 6.58 nm (see Suppl. Fig. 5).

The energy-dispersive X-ray spectroscopy (EDS) reveals that all the elements (N, Pb, and Br) are uniformly distributed on the whole pattern (Fig. 3g). To verify the monocrystallity of the printed MAPbBr$_3$ structure, we used an 805 nm femtosecond laser to excite two-photon fluorescence and recorded the intensity at different incident polarizations[39]. We can see that the polarization-dependent fluorescence intensity distribution shows quartic symmetry, which is consistent with the lattice symmetry of cubic-phase MAPbBr$_3$. The similar results recorded at the four locations (I−IV), which are spatially separated reveal that the crystal orientations at these locations are the same (Fig. 3h). We further verified the single-crystal structure by micro X-ray diffraction. As shown in Fig. 3i, we observed narrow and sharp diffraction peaks arising from (00$l$) surface of the cubic-phase MAPbBr$_3$, suggesting a highly oriented single crystal. Moreover, we examined the Raman spectrum of the printed patterns, where the symmetric and asymmetric stretching modes of Pb-Br bonds, the torsion mode of the MA group, the symmetric bending mode of -CH$_3$ and -NH$_3^+$, and the asymmetric bending mode of -NH$_3^+$ are clearly identified (see Suppl. Fig. 6).

We also evaluated the key performance of the printed halide perovskite materials for optical or optoelectronic applications. The MAPbBr$_3$ single crystal exhibits a clear band-edge absorption without observation of any additional excitonic signal, indicating a minimal number of in-gap defects (see Suppl. Fig. 7). The PL spectrum shows a narrow emission peak at 539 nm with a full width at half-maximum

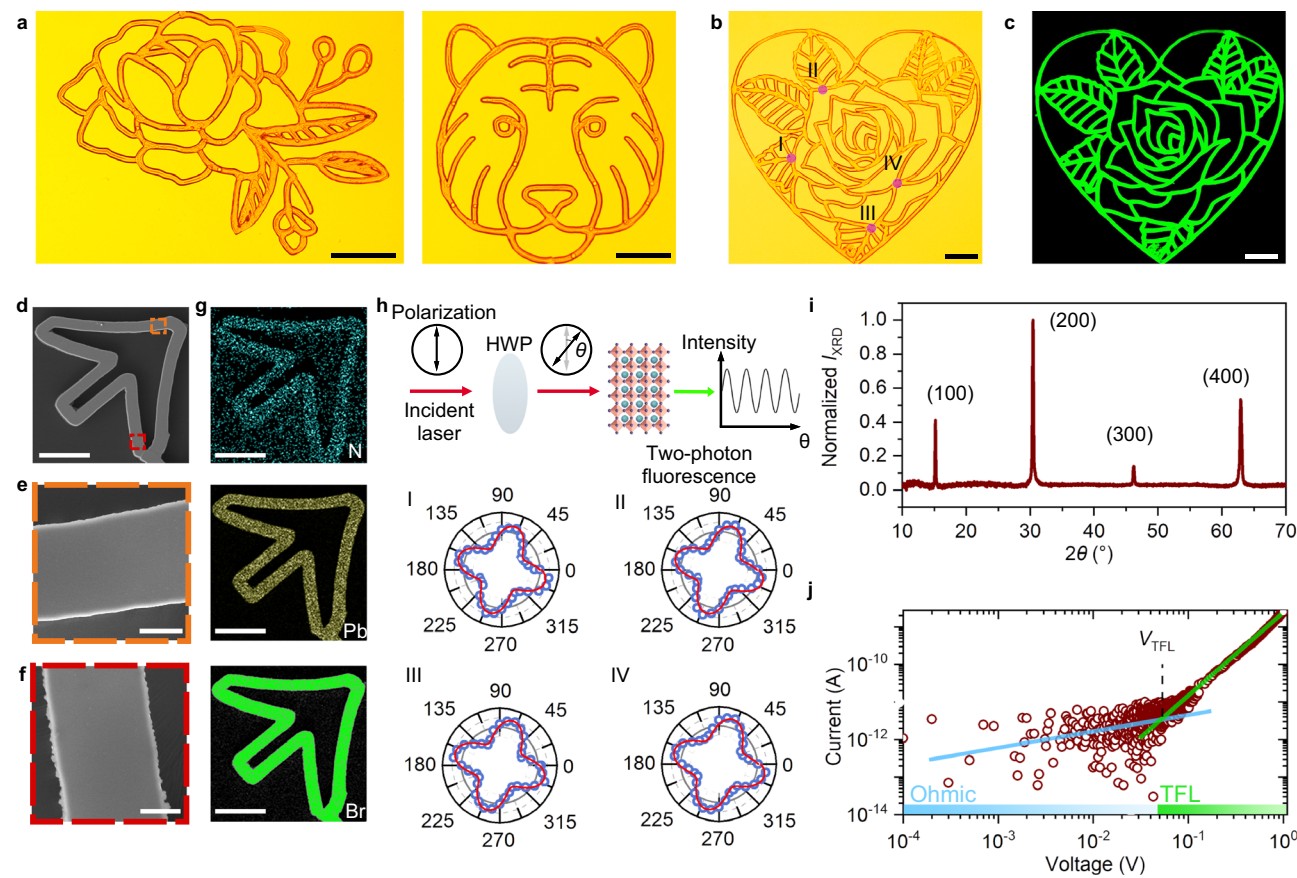

**Fig. 3 | The fabrication and characterization of diverse MAPbBr$_3$ single crystals.**
**a** The optical images of different MAPbBr$_3$ single crystals. Scale bar: 100 μm.
**b**, **c** The optical image and the corresponding confocal fluorescence image of MAPbBr$_3$ single crystalline pattern. Scale bar: 100 μm. **d**–**f** The SEM images of an arrow-shaped single crystal. **e** and **f** show the close-up images of the marked area in (**d**). Scale bar: 50 μm for (**d**) and 5 μm for (**e**) and (**f**). **g** The corresponding EDS mapping results for the structure in (**d**). Scale bar: 50 μm. **h** Polarization-dependent two-photon fluorescence intensity recorded at the four different locations marked as I–IV in the bottom panel of (**b**). The principle of the optical characterization is schematically illustrated at the top panel. **i** Micro X-ray diffraction spectrum of the printed MAPbBr$_3$ single-crystal structure. **j** SCLC characterization of the MAPbBr$_3$ structure. The linear Ohmic regime is marked by a blue line, while the trap-filled regime is marked by a green line. The voltage threshold is given as $V_{TFL}$. Source data are provided as a Source Data file.

(FWHM) of 20 nm. A biexponential fit of the time-resolved PL spectrum reveals a superposition of fast (surface component) and slow (bulk component) decay dynamics, with lifetimes of 1.6 ns and 30.2 ns, respectively. The lifetimes are comparable to that of spontaneously grown crystals (Suppl. Fig. 8a), indicating that the laser irradiation did not introduce additional structural defects. To quantitatively evaluate the density of electronic trap states in the printed materials, we carried out space charge limited current (SCLC) measurement[27,40]. As shown in Fig. 3j, the dark current−voltage curve depicts a clear transition from the Ohmic region to the trap-filled limit (TFL) region at $V_{TFL} = 0.052$ V. The trap density can then be calculated by

$$n = 2\,\varepsilon\,\varepsilon_0\,V_{TFL}\,/\,(e\,d)^2 \tag{1}$$

where $e$ is the elementary charge, $d$ is the channel length (20 μm in our device), while $\varepsilon$ and $\varepsilon_0$ are the relative and vacuum permittivity, respectively. We obtain a trap density of $3.7 \times 10^{11}$ cm$^{-3}$ in the printed MAPbBr$_3$ single crystals, which is 3−4 orders of magnitude lower than that of polycrystalline thin films[41].

## General applicability of OCL

Beyond MAPbBr$_3$, we further demonstrate that the OCL strategy can be extended to other halide perovskites since the same principle works generally. Firstly, we replaced MABr with formamidine bromide (FABr) in the precursor solution to fabricate FAPbBr$_3$ single crystal structure. It is noted that surface ligands should be selected according to the composition of halide perovskites. For FAPbBr$_3$, the mixture of oleic acid and oleylamine is not an optimized option as it leads to a thin and fragile crystal structure. Instead, taking octylamine as the surface ligand, the spontaneous growth can be significantly suppressed without performance deterioration. We also demonstrate that OCL can be applied to fabricate single-crystal halide perovskites composed of various cations, such as Cl$^-$, I$^-$, and so on. To print the MAPbCl$_3$ single crystals, we chose a mixed solvent of DMF and DMSO with a stoichiometric ratio of 1:1 to improve the solubility. For MAPbI$_3$, hydroiodic acid was employed as solvent and the precursor solution was heated to 321 K before laser printing (see Methods).

The various halide perovskite single crystals printed by OCL are summarized in Fig. 4a−i. All the target elements are uniformly distributed among the patterns. The morphologies and crystal structures are verified by SEM and Raman spectroscopy, respectively (see Fig. 4j and Suppl. Table 1 for detailed analysis). All the printed materials exhibit excellent PL performance, with the cation-tunable emission wavelength of 404 nm, 550 nm, and 773 nm, and the FWHM of 12 nm, 20 nm, and 42 nm for MAPbCl$_3$, FAPbBr$_3$, and MAPbI$_3$, respectively (Fig. 4k). The absorption spectra reveal the absence of the excitonic signal, while the fluorescence lifetimes summarized in Fig. 4l elucidate the low defect densities in all the samples (also see Suppl. Fig. 8b−d for the lifetime of the spontaneously grown samples for comparison). In addition to organic-inorganic hybrid perovskites, we also verified that OCL can be extended to all-inorganic perovskites (see Suppl. Fig. 9 the fabrication of cesium lead bromide micropattern).

Harnessing the coordinated effect of optical manipulation of molecular species in the precursor solution and laser-controlled adsorption-desorption trade-off of surface ligands, we develop the OCL technique to control the ionic reaction and growth kinetics in crystallization. With its precise spatiotemporal control of the local supersaturation and surface energy at the growth interface, OCL allows direct printing of arbitrary halide perovskite single-crystal micropatterns of high surface quality, crystallinity, and low trap state density. Specifically, it significantly averts external chemical damage to the delicate halide perovskite crystals, which easily occurs in traditional lithography process. Considering the working principle, the OCL

approach can be potentially extended to the fabrication of single-crystal structures of other materials based on supersaturation-driven crystallization, if highly volatile solvent can be employed to dissolve the precursor solute and the grown materials own high optothermal conversion efficiency.

## Methods

### Chemicals and materials

Lead (II) bromide (PbBr$_2$, Adamas, 99%), lead (II) iodide (PbI$_2$, TCI, 99%), lead (II) chloride (PbCl$_2$, TCI, 99%), hydriodic acid (HI, Aladdin, 55-58%), methylammonium iodide (MAI, Meryer, 99.5%), methylammonium bromide (MABr, Meryer 99.5%), methylammonium chloride (MACl, Meryer, 99.5%), formamidine bromide (FABr, P-OLED(Shanghai) Technology CO., LTO, 99.5%), oleic acid (OA, Thermo scientific, 90%), oleylamine (OAm, Thermo scientific, 80-90%), octylamine (Meryer, 99%). Isopropanol, alcohol, *N,N*-dimethylformamide (DMF 99%), and dimethyl sulfoxide (DMSO 99%) were purchased from general reagents. All reagents were employed as received without further purification. The saturation concentration of different halide perovskites is obtained from refs. 26,42−45.

### Precursor preparation of MAPbBr$_3$

0.4980 g PbBr$_2$ and 0.1519 g MABr (molar ratio 1:1) were mixed in 1 mL DMF and stirred for 20 min to get the saturated MAPbBr$_3$ precursor without ligand[26]. 20 μL OA and 60 μL OAm were mixed and stirred for 10 min to get the mixed ligand solution. Then 0.67 μL mixed ligand solution was added into 150 μL MAPbBr$_3$ precursor to get the MAPbBr$_3$ precursor with ligand (volume ratio of OAm to ligand-free precursor was 1:300). In all precursor solutions used, the volume ratio of OA to OAm was 1:3. The precursor solution (1: $x$) represents the volume ratio of the OAm to the ligand-free precursor solution.

### Precursor preparation of MAPbI$_3$

0.463 g PbI$_2$ and 0.159 g MAI (molar ratio 1:1) were mixed in 2 mL HI (52%) and stirred for 20 min to get the saturated MAPbI$_3$ precursor without ligand[42]. Then 0.5 μL OA and 0.5 μL OAm were added in 200 μL MAPbI$_3$ precursor to get the MAPbI$_3$ precursor with surface ligand (volume ratio of OAm to ligand-free precursor was 1:400). After the addition of ligands, the solution was stirred for more than 10 min. At this time, precipitation would occur in the solution, and the supernatant was selected for further experiments.

### Precursor preparation of MAPbCl$_3$

0.0742 g MACl and 0.3057 g PbCl$_2$ (molar ratio 1:1) were mixed in 1 mL DMF/DMSO (volume ratio 1:1) and stirred for 20 min to get the saturated MAPbCl$_3$ precursor without ligand[43]. 20 μL OA and 60 μL OAm were mixed and stirred for 10 min to get the mixed ligand solution. Then 0.5 μL mixed ligand solution was added into 100 μL MAPbCl$_3$ precursor to get the MAPbCl$_3$ precursor with ligand (volume ratio of OAm to ligand-free precursor was 1:200).

### Precursor preparation of FAPbBr$_3$

0.463 g PbBr$_2$ and 0.159 g FABr (molar ratio 1:1) were dissolved in 1 mL DMF and stirred for 20 min to get the saturated FAPbBr$_3$ precursor without ligand[44]. Then 1 μL octylamine was added in 50 μL FAPbBr$_3$ precursor to get the FAPbBr$_3$ precursor with ligand (volume ratio of octylamine to ligand-free precursor was 1:50).

### Precursor preparation of CsPbBr$_3$

0.202 g CsBr and 0.697 g PbBr$_2$ (molar ratio 1:2) were dissolved in 1 mL DMSO and stirred for 20 min to get the saturated CsPbBr$_3$ precursor without ligand[45]. Then 1 μL octylamine was added in 80 μL CsPbBr$_3$ precursor to get CsPbBr$_3$ precursor with ligand (volume ratio of octylamine to ligand-free precursor was 1:80).

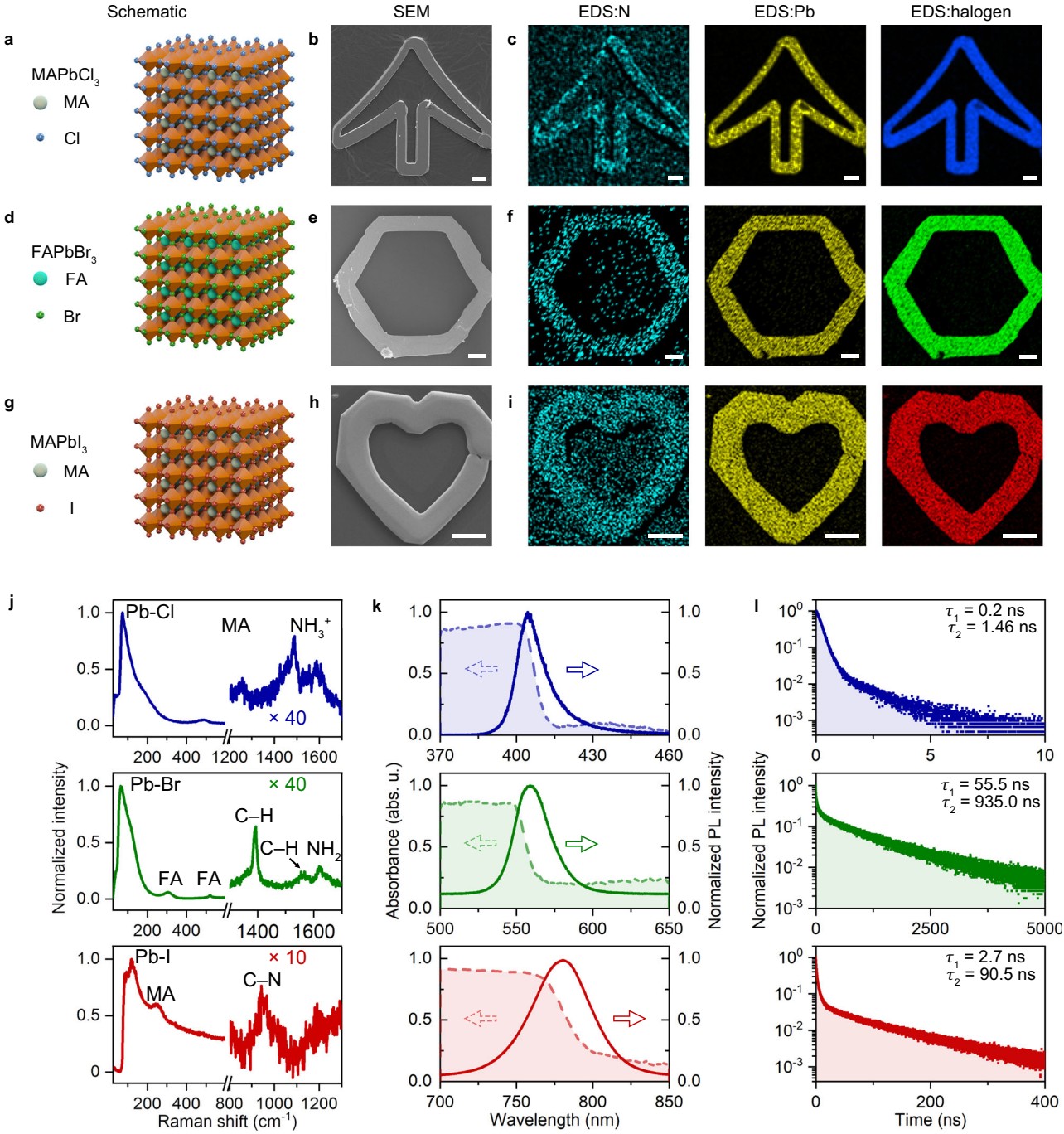

**Fig. 4 | General applicability of OCL. a** Lattice structure, **b** SEM, and **c** EDS mapping of a MAPbCl₃ arrow micropattern. **d** Lattice structure, **e** SEM, and **f** EDS mapping of a FAPbBr₃ hexagon micropattern. **g** Lattice structure, **h** SEM, and **i** EDS mapping of a MAPbI₃ heart micropattern. Scale bar: 20 µm. **j** Raman spectra of the different halide perovskite structures. **k** Absorption and PL spectra of the different halide perovskite structures. **l** Time-resolved PL spectra of the different halide perovskite structures. Source data are provided as a Source Data file.

## Laser direct printing of halide perovskites using OCL

All experiments (except MAPbCl₃) were performed using a femtosecond laser (FemtoYL-10, Wuhan Yangtze Soton Laser Co., Ltd.) operating at 1030 nm integrated with Nikon Ti2-u microscope (Suppl. Fig. 10). 515 nm femtosecond laser was generated by second-harmonic generation from the 1030 nm femtosecond laser, and further integrated in Ti2-u microscope to trigger OCL. The microscope was equipped with a 20× objective lens (NA = 0.5, WD = 2.1, Nikon) to focus the laser on the substrate. The glass substrates were sequentially washed with isopropanol, alcohol, and deionized water. A spacer (Thermo Fisher, S24737, 9 mm in diameter) was pasted on the substrate to limit the precursor, then the substrate was further treated with oxygen plasma (Saot Technology, 90 W, air, 2 min). The precursor (usually 0.8 µL) was dropped in the middle of the spacer and then was sealed in a homemade chamber of 0.5 mm thickness. Motorized positioning systems (Marzhauser Wetzlar) were used to control the movement of the laser spot. Due to the spontaneous evaporation of the saturated precursor solution, spontaneous nucleation may occur and provide seed crystals to start the OCL process. In addition, seed crystals can also be created by irradiation of a high-power (e.g., 15 mW) femtosecond laser, which generates a microbubble to concentrate the precursor solute for nucleation. The irradiation time should be

carefully controlled to avoid laser damage to the crystals. To enable the laser direct printing, the motorized positioning systems were programmed in a certain path. By steering the laser spot location, the perovskite structures were induced to a set of patterning. For MAPbI$_3$, a heater integrated with a microscope was used to maintain the temperature at 321 K. A 405 nm picosecond laser (PicoQuant PDL 800-D pulsed diode laser) was used to fabricate MAPbCl$_3$ structure. For CsPbBr$_3$, a heater integrated with a microscope was used to maintain the temperature at 355 K. After crystallization, the perovskite crystals were still immersed in the precursor solution. The crystals are not dissolved because the bulk precursor solution is saturated. We use octane to separate the grown structures from the precursor as it cannot dissolve the perovskite crystals and it is not miscible with the solvent in the precursor to suppress nucleation during the processing.

## Optical characterizations
The temperature distribution was recorded by Phasics SID 4 camera integrated into a Nikon Ti2-U microscope. A 515 nm femtosecond laser of different optical powers was focused on the perovskite crystal. The phase change of the white light was recorded in situ by Phasics SID4 camera to measure the refractive index change of the solvent, and the temperature variation is then derived via an inversion problem algorithm as the relation between the refractive index and the temperature is known[46]. The absorption spectra of different halide perovskite materials were measured by Andor Shamrock 500i integrated into a Nikon Ti2-u microscope. The PL spectra and the Raman spectra of different perovskites were carried out using a high-resolution Raman spectrometer (HR800, Horiba). The wavelength of the exciting laser for PL spectrum of MAPbCl$_3$ was 325 nm, while for other perovskites (MAPbBr$_3$, MAPbI$_3$, and FAPbBr$_3$) the wavelength was 473 nm. The excited laser for Raman spectrometry of MAPbCl$_3$, MAPbBr$_3$, and FAPbBr$_3$ was 633 nm. 532 nm laser was used to get the Raman spectrum of MAPbI$_3$ under N$_2$ atmosphere. The confocal images were recorded with a Nikon A1 confocal microscope (Nikon Eclipse Ti2-E, objective lens Nikon Plan Fluro 40×/0.6) and excited by a 488 nm continuous-wave laser.

The crystal orientation characterization was performed using an 805 nm femtosecond laser (Mai Tai) integrated with a Nikon A1 confocal microscope (Nikon Eclipse Ti2-E, objective lens Nikon Plan Fluro 40×/0.6). The setup is shown in Suppl. Fig. 11. A Glan polarizer and a piece of half-wave plate were used to control the polarization direction of the femtosecond laser. Then the laser was focused on the sample to excite two-photon fluorescence. The fluorescence light was filtered (Thorlabs, 700 nm, shortpass) and then collected by a spectrometer (Andor KYMERA-328I-B1, grating 300 grooves mm$^{-1}$). The intensity was normalized by incident power.

## Optical tracking and analysis
To reveal the Marangoni convection flow, polystyrene microspheres (Macklin 5% w/v, diameter 2 μm) were used for optical tracking. The precursor for tracking was prepared by adding proper polystyrene microspheres into the ligand-free MAPbBr$_3$ precursor followed with ultrasonic processing. The optical images of OCL-induced growth by using the precursor for tracking were recorded by Andor Neo 5.5 sCMOS. The trajectories of polystyrene microspheres during growth were analyzed by Imaris 9.3.1.

## Scanning electron microscopy
The SEM images and EDS images were recorded using Zeiss field-emission scanning electron microscope instrument (Merlin) and Hitachi cold field-emission scanning electron microscope instrument (SU-8010). The accelerating voltage was set to 10 kV.

## X-ray diffraction measurement
A Bruker D8 Discover micro-focal spot two-dimensional X-ray diffractometer system was used for the micro X-ray diffraction measurement, with a copper focus X-ray tube (K$_\alpha$: 1.54 Å). The diameter of spot was 1 mm. Measurements were conducted in air.

## Time-resolved photoluminescence (TRPL) measurement
TRPL of MAPbBr$_3$, MAPbI$_3$, and FAPbBr$_3$ was measured by a home-built setup. A 405 nm pulsed femtosecond laser (Mai Tai) with a repetition rate of 4 MHz was focused onto the surface of the sample using an objective lens (Olympus, 20×, SApo, NA = 0.7), generating a Gaussian-distribution spot with a FWHM of 1366 nm (for FAPbBr$_3$, the repetition rate was 0.5 MHz). All measurements were done with an average incident power of ≈40 nW. Then, the PL from the photoexcitation, together with the reflected and scattered excitation signal, was collected by the same objective lens. The resulting collimated beam passed through a long-pass filter with a cutoff at $\lambda$ = 425 nm to remove the reflected and scattered excitation light. The resulting PL signal was further collected through a multimode fiber (Thorlabs, 50 μm in diameter, 5 m) with a pair of coupler and collimator (Thorlabs RC08FC-P01) and sent to an APD (MPD PDM Series 20 μm), which was attached to the timing electronics (PicoQuant PicoHarp 300).

The TRPL of MAPbCl$_3$ was measured by a home-built setup. 515 nm femtosecond laser was used to excite PL signal by two-photon absorption. A 450 nm short pass filter was used to remove the reflected and scattered excitation light. The resulting PL signal was further collected and sent to an APD (MPD PDM Series 20 μm) which is attached to the timing electronics (PicoQuant PicoHarp 300).

## Electrical characterizations
The MAPbBr$_3$ structure was transferred by polydimethylsiloxane on a commercially available interdigitated array electrode (Mecart Sensor Tech, line distance: 20 μm) to prepare the sample for electrical characterization. Current as a function of the applied voltage was measured by semiconductor parameter analyzer (Agilent B1500A). The sample was kept in a dark environment. A non-linear response was observed and analyzed according to space charge limited current theory.

## Numerical simulation
A finite element solver (COMSOL Multiphysics 5.4) was used to simulate the temperature profile and concentration profile of the setup. The physics involved included fluid dynamics in laminar flow (DMF), heat transfer in solid and liquid (DMF and halide perovskite crystal), and transport of concentrated species (DMF-perovskite precursor). A coupling of multi-physics, including non-isothermal flow, reaction flow, and the Marangoni effect, was considered in this model. In the simulation, the incident Gaussian beam was launched and irradiated on a perovskite crystal immersed in a saturated DMF-perovskite precursor. The boundary condition for heat transfer physics was convection heat exchange at the liquid-gas and liquid-solid interface. The physics model involved heat transfer in solids and fluids. The Multiphysics coupling included boundary heat source, temperature coupling, non-isothermal flow, and the Marangoni effect. Boundary conditions for the heat transfer were set as scattering boundary and room temperature. For the laminar flow physics, the liquid-gas interface was set as a slip interior wall, while the liquid-solid interface was set as nonslip walls. An open boundary condition was considered for heat transfer, laminar flow, and mass diffusion at the lateral boundaries. The thermodynamic parameters of MAPbBr$_3$ and DMF were taken from refs. 36,47,48.

## Reporting summary
Further information on research design is available in the Nature Portfolio Reporting Summary linked to this article.

## Data availability

The data that support the findings of this study are available from the corresponding author upon request. Source data are provided in this paper.

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

## Acknowledgements

The authors acknowledge the financial support of the National Key Research and Development Program of China (2020YFA0715000, L.L.), the Tsinghua University Initiative Scientific Research Program, the National Natural Science Foundation of China (62075111, L.L. and 61960206003, H.-B.S.), Tsinghua-Foshan Innovation Special Fund (2021THFS0102, H.-B. Sun), Tsinghua-Deqing Joint Research Center for Materials Design and Industrial Innovation, and the State Key Laboratory of Precision Measurement Technology and Instruments. X.-G.C. thanks the Shanghai Institute of Ceramics, Chinese Academy of Science, for providing professional help in micro X-ray diffraction characterization. X.-G.C. thanks B. Liu for helping analyze the tracking experiment. We acknowledge the support from the Nano-optics laboratory as well as other characterization platforms.

## Author contributions

X.-G.C., L.L. and H.-B.S. conceived the idea. L.L., H.-B.S. and Z.L. supervised the work. X.C. designed and conducted the fabrication and characterization experiments. G.-Y.H. and T.F. contributed the simulation results. X.-M.C. provided the optimized conditions to fabricate $MAPbI_3$. X.-Z.L. and Y.-K.Z. performed the experimental work on the TRPL measurements. Y.Z. and P.L. conducted the dark current measurement. All authors participated in the discussion of the results and wrote the paper.

## Competing interests

The authors declare no competing interests.
