## [Peer Review File · Nature Communications]

Optofluidic crystallithography for directed growth of single-crystalline halide perovskiteREVIEWER COMMENTS

Reviewer #1 (Remarks to the Author):

The manuscript reports a nice creative technique to deposit single crystalline or polycrystalline halide perovskite pattern from solution using local heating by a laser beam. The deposited patterns are well characterized and exhibit good quality. The method is technically simple and deserves attention from the scientific community. I would recommend publication in Nature Communications after a revision.

My main concern is that despite all discussions some simple basic idea behind this technique is not clearly formulated. I understand that local supersaturation is created by evaporation of solution due to a local heating. The material supplied to the crystallization front by convection. After crystallization solution is depleted with the solvent preventing dissolution of crystallized material. Is it correct?

What happens to solution after crystallization? How dense (crystallized area with respect to total area) can be this pattern? How solution layer thickness affects crystallization? How consistent is thickness of the deposited crystalline pattern?

The manuscript contains a lot of non-standard terminology such as "post-nucleation growth kinetics", "growth forefront", "growth rate is dramatically improved", "improvement of local precursor" etc. Moreover, the title of the manuscript "Optofluidic Crystalithography" does not make too much sense.

Reviewer #2 (Remarks to the Author):

The manuscript of Chen et al provides an elegant mechanism to control the crystal growth of perovskite crystals using light. When the issues and topics below are clarified, this manuscript could fit well in Nature Communications.

1. Introduction on light-induced crystallization and perovskites: There is a rapidly growing body of work on controlling crystallization processes with light. Similarly control of crystallization of perovskites and their material properties is well-studied. The manuscript contains hardly any background on these two topics in the introduction. It would help the reader to include a short intro on the state-of-the-art such that the current work is placed in perspective.
2. Mechanistic details: the main text is rather vague about the mechanism, whereas the SI contains quite some insights. I would suggest moving part of this discussion to the main text.
3. The claim on generality is overstated: The authors demonstrate the principle for three perovskite compounds, but the mechanism is likely not working for any compound. It would help to be clear about the limitations, which can also be combined with the current insights on the mechanism.
4. Evidence for claim on surface quality: The authors frame the work around controlling defect density, morphology, and high surface quality. The control on growth at macroscopic scale is very clearly demonstrated, and impressive, but evidence for the quality on the surface quality etc. is rather minimal. Only one SEM in the SI shows macroscopic roughness, and Figure 3c shows only a zoomed-out SEM micrograph which makes it hard to judge the surface roughness. It would therefore be important to include a close-up SEM or AFM in the main text to show how smooth this surface is.
5. Experimental details: Some essential details are not clear/missing from the main text or SI. Please incorporate these details:
 - i) How is the temperature measured in/near the laser spot.
 - ii) How is concentration of the precursors selected. Is this done in simulations or experimentally? Is the temperature measured in the presence with the spot focused on a crystal and next to a crystal? Is the solution already initially supersaturated?
 - iii) One sentence is mentioned on the nucleation part, but this remains very unclear. It would be helpful to incorporate details on how you start the process, especially because the current mechanism

can not explain exactly how the seed crystal forms.

iv) It would be helpful to provide details on the focal spot and power density in the main text. Also, the pulsed laser likely has much higher peak power, so a comparison between both illumination systems should be done carefully. For instance, does the crystal surface get damaged during the pulsed laser shots?

v) The surfactant addition is very elegant. Do you also need to increase the laser power to induce nucleation and maintain growth?

vi) Line 82 "The local solute concentration can be further tuned by the laser power" This is not clear. Do you mean that the solution evaporates faster?

vii)Line 119: "solubility is reduced at an improved temperature" I assume the authors mean an increased temperature. Please provide a reference or data that show that this is indeed the case.

Reviewer #3 (Remarks to the Author):

Response to reviewer's comments

Referee 1

Comments:

The manuscript reports a nice creative technique to deposit single crystalline or polycrystalline halide perovskite pattern from solution using local heating by a laser beam. The deposited patterns are well characterized and exhibit good quality. The method is technically simple and deserves attention from the scientific community. I would recommend publication in Nature Communications after a revision.

Answer: Thank you very much for the reviewer's positive and insightful comments.

We have made point-to-point responses to the comments:

1. My main concern is that despite all discussions some simple basic idea behind this technique is not clearly formulated. I understand that local supersaturation is created by evaporation of solution due to a local heating. The material supplied to the crystallization front by convection. After crystallization solution is depleted with the solvent preventing dissolution of crystallized material. Is it correct?

Answer: Thank you very much for the reviewer's comments. As pointed out by the reviewer, the local evaporation is the key factor for the formation of supersaturation environment as it reduces the amount of solvent molecules and improves the solute concentration. The evaporation rate can be calculated from the Hertz-Knudsen equation:

$$J = \frac{p}{\sqrt{2\pi mk_B T}}$$

where J is the flux that molecules evaporating from solvent to air, m is the molecular

weight, k_B is the Boltzmann's constant, T is the temperature of the surface, and p is the corresponding saturated vapor pressure. Usually, the evaporation rate is improved at higher temperature because p is more sensitive to temperature T , which can be described by Wagner equation:

$$\ln\left(\frac{p}{p_c}\right) = (A_1\tau + A_2\tau^{1.25} + A_3\tau^3 + A_4\tau^7) \frac{T_c}{T}$$

where $\tau=1-T/T_c$, T_c is the critical temperature and p_c is the critical pressure. A_n ($n=1,2,3,4$) is the coefficients which can be determined in the experiments. For DMF, $T_c=596.6$ K and $p_c=5.220$ MPa. A_n can be obtained in Ref. 1.¹ Consequently, the local supersaturation can be improved by increasing the optical power to increase the local temperature, which accelerates the local evaporation.

After the laser-directed crystallization, the perovskite crystals were still immersed in the precursor solution. The crystals are not dissolved because the bulk precursor solution is saturated. Instead, without any surface engineering, spontaneous crystal growth is observed (as revealed in Fig. 2b in the manuscript). The precursor solution is removed using octane, which rapidly separates the grown structures to extract the samples for further characterization. Octane was chosen as it cannot dissolve the perovskite crystals and it is not miscible with the solvent in the precursor to suppress nucleation during the processing.

According to the reviewer's suggestion, we have added more discussion on the working mechanism in both the main text and the Supplementary Materials

Main text, lines 6-13, page 5:

“The local temperature gradient drives thermal diffusion, Marangoni convection, and

interfacial evaporation in the precursor solution, which redistribute the molecular species around the laser spot (see Fig. 1g and detailed analysis in Supplementary Note 1).” *was replaced by* “As the thickness of precursor solution layer is thin, the local optical heating introduces temperature gradient at the liquid/gas interface and drives Marangoni convection (see Fig. 1g) and interfacial evaporation. It is noted that the local evaporation reduces the amount of solvent molecules and improve the solute concentration, which is the key factor for the formation of supersaturation environment. The evaporation rate increased with the local temperature due to higher saturated vapor pressure. The Marangoni convection, together with the non-isothermal diffusion, redistribute the molecular species in the solution and creates a local supersaturation environment around the laser spot (Supplementary Note 1).”.

Methods, lines 22-23, page 23 and lines 1-2, page 24:

“After crystallization, the perovskite crystals were still immersed in the precursor solution. The crystals are not dissolved because the bulk precursor solution is saturated. We use octane to separate the grown structures from the precursor as it cannot dissolve the perovskite crystals and it is not miscible with the solvent in the precursor to suppress nucleation during the processing.” *was added*.

Supplementary Information, lines 177-178, page 13 and lines 179-189, page 14:

“The evaporation rate can be calculated from the Hertz-Knudsen equation:

$$J = \frac{p}{\sqrt{2\pi mk_B T}}$$

In this equation, J is the flux that molecules evaporating from solvent to air, m is the molecular weight, k_B is the Boltzmann's constant, T is the temperature of the surface,

and p is the corresponding saturated vapor pressure.

Usually, the evaporation rate is improved at higher temperature because p is more sensitive to temperature T , which can be described by Wagner equation:

$$\ln\left(\frac{p}{p_c}\right) = (A_1\tau + A_2\tau^{1.25} + A_3\tau^3 + A_4\tau^7) \frac{T_c}{T}$$

Where $\tau = 1 - T/T_c$, T_c is the critical temperature and p_c is the critical pressure. A_n ($n=1,2,3,4$) is the coefficients which can be determined in the experiments. For DMF, $T_c = 596.6$ K and $p_c = 5.220$ MPa. A_n can be obtained in Ref. 10.¹⁰ The calculated J was used for further COMSOL simulation. The reduction of solute molecules due to evaporation at the interface leads to significant supersaturation at the laser spot.” *was added.*

Ref. 1 *was added* as Ref. 10 in the supplementary information.¹

2. What happens to solution after crystallization? How dense (crystallized area with respect to total area) can be this pattern? How solution layer thickness affects crystallization? How consistent is thickness of the deposited crystalline pattern?

Answer: Thank you very much for the reviewer’s insightful comments. As discussed in Response to Comment 1, after crystallization, the structures are still immersed in the precursor solution. As the precursor solution is saturated, dissolution is not observed. Instead, spontaneous crystallization can easily occur without surface ligand protection. The precursor solution is removed from the substrate using octane, which rapidly separates the precursor solution from the grown structures.

For the micropatterns created in Fig. 3 in the main text, the filling factor (considering the pattern area) ranges from 17.1% to 27.2%. The filling factor f is calculated by:

$$f = \frac{a_p}{a_s} \times 100\%$$

a_p is the area of perovskite micropatterns and a_s is the area of the minimum rectangle encompassing the micropatterns (as indicated by the dash rectangle in Fig. R1).

Figure R1. The calculation of filling factor. The area of dash rectangle which encompasses the “rose” pattern is defined as a_s .

In this work, we used a focus laser spot to define the crystal growth pathway, and the filling factor relies on the design of the micropatterns and the patterning linewidth. In principle, a filling factor of 100% can be achieved when the laser scanning pathway is designed to grow a continuous perovskite thin film. A more feasible strategy is to use a line-shaped laser beam to direct the growth kinetics. As shown in Fig. R2, we used a concave cylindrical lens and a convex cylindrical lens to create a line pattern of 2 μm in width and 180 μm in length (after a 20 \times objective). A scanning of the line-shaped laser beam can rapidly create large-area and dense perovskite thin film. For demonstration, we fabricated a MAPbI_3 thin film of 180 $\mu\text{m} \times 5 \text{ mm}$ in area. Such

fabrication process can be potentially scaled up for fabrication of large-area single-crystal perovskite thin film.

Figure R2. Optical printing of halide perovskites using a line-shaped laser beam.

a, Schematic of the optical setup. A set of concave and convex cylindrical lenses were used to create a line pattern of 2 μm in width and 180 μm in length (after a 20 \times objective). **b**, The optical images of line-shaped laser beam and MAPbI_3 thin film created. A heater was used to maintain the processing temperature at 80 $^\circ\text{C}$. Scale bar: 50 μm .

Considering the working principle of optofluidic crystallography, the thickness of precursor solution is critical as it determines the local evaporation rate at the laser spot and the formation of local supersaturation. The increase of precursor thickness reduces the temperature gradient at the liquid-gas interface and strongly suppress the convective flow velocity. Thus, the local evaporation rate is reduced and the precursor concentration cannot be significantly improved. In our experiments, the typical solution

thickness is 8-15 μm . The laser-directed growth of halide perovskites cannot be observed when the solution thickness is above 30 μm .

The grown single crystal micropattern of halide perovskites through optofluidic crystallography has a uniform thickness distribution. As shown in Fig. R3, we used white light interferometry to measure the thickness distribution of a grown arrow pattern of 150 $\mu\text{m} \times 150 \mu\text{m}$ in area. We can see that the thickness across the pattern surface is uniform.

Figure R3. White light interferometry image of the structure. Scale bar: 10 μm

According to the reviewer's suggestion, we have provided more details and discussion in the Methods section and the main text:

Methods, lines 22-23, page 23 and lines 1-2, page 24:

“After crystallization, the perovskite crystals were still immersed in the precursor solution. The crystals are not dissolved because the bulk precursor solution is saturated. We use octane to separate the grown structures from the precursor as it cannot dissolve perovskite and it is not miscible with perovskite solvents to suppress nucleation during the processing.” *was added.*

Main text, lines 4-11, page 13:

“For the micropatterns created in Fig. 3, the filling factor ranges from 17.1% to 27.2% for a single pattern. In principle, a filling factor of 100% can be achieved when the laser scanning pathway is designed to grow a continuous perovskite thin film. A more feasible strategy is to use a line patterned laser beam to direct the growth kinetics. As shown in Supplementary Fig. 4, we used a concave cylindrical lens and a convex cylindrical lens to create a line pattern of 2 μm in width and 180 μm in length (after a $20\times$ objective). A scanning of the line laser pattern can rapidly create large-area and dense perovskite thin film.” *was added*.

Main text, lines 21-24, page 4:

“As shown in Fig. 1a, a layer of saturated MAPbBr_3 precursor solution, i.e., the mixture of equivalently stoichiometric methylammonium bromide (MABr) and lead bromide (PbBr_2) dissolved in N-N dimethylformamide (DMF), is placed inside a chamber on a glass substrate.” *was replaced by* “As shown in Fig. 1a, a layer of saturated MAPbBr_3 precursor solution of 15 μm in thickness, i.e., the mixture of equivalently stoichiometric methylammonium bromide (MABr) and lead bromide (PbBr_2) dissolved in N-N dimethylformamide (DMF), is placed inside a chamber on a glass substrate.”.

Main text, lines 15-22, page 8:

“It is noted that the liquid-gas interface is also indispensable here as it provides an extraction channel to deliver the solvent molecules towards the interface for evaporation, which facilitates the improvement of solution concentration (see Supplementary Note 6).” *was replaced by* “It is noted that the thickness of precursor

solution is critical as it determines the local evaporation rate at the laser spot and the formation of local supersaturation (see Supplementary Note 6). The increase of precursor thickness reduces the temperature gradient at the liquid-gas interface and strongly suppress the convective flow velocity. Thus, the local evaporation rate is reduced and the precursor concentration cannot be significantly improved. In our experiments, the typical solution thickness is 8-15 μm . The laser-directed growth of halide perovskites cannot be observed when the solution thickness is above 30 μm .”.

Supplementary Information, lines 51-56, page 4:

Fig. R2 *was added* in the Supplementary Materials as Supplementary Fig. 4.

3. The manuscript contains a lot of non-standard terminology such as “post-nucleation growth kinetics”, “growth forefront”, “growth rate is dramatically improved”, “improvement of local precursor” etc. Moreover, the title of the manuscript “Optofluidic Crystalithography” does not make too much sense.

Answer: Thank you very much for the reviewer’s comments. We have changed the non-standard terminology in the manuscript.

“post-nucleation growth kinetics” *was replaced by* “growth kinetics”.

“growth forefront” *was replaced by* “growth interface”.

“improvement of local precursor concentration” *was replaced by* “improvement of local supersaturation”.

We have also revised the title for the manuscript:

“Optofluidic Crystalithography” *was replaced by* “Optofluidic Crystalithography for

directed growth of single-crystal halide perovskites”.

Referee 2

Comments:

The manuscript of Chen et al provides an elegant mechanism to control the crystal growth of perovskite crystals using light. When the issues and topics below are clarified, this manuscript could fit well in Nature Communications.

Answer: Thank you very much for the reviewer's positive and insightful comments.

We have made point-to-point responses to the reviewer's comments:

1. Introduction on light-induced crystallization and perovskites: There is a rapidly growing body of work on controlling crystallization processes with light. Similarly control of crystallization of perovskites and their material properties is well-studied. The manuscript contains hardly any background on these two topics in the introduction. It would help the reader to include a short intro on the state-of-the-art such that the current work is placed in perspective.

Answer: Thank you very much for the reviewer's suggestion. Optical control of crystallization process has been widely studied and different approaches have been developed so far. For example, laser has been used to control the crystallization of proteins by generating microbubbles or concentrating protein molecules in the optical potential.^{2,3} Laser-induced non-photochemical nucleation has also been observed in various salts and amino acids.⁴ As pointed out by the reviewer, laser control of crystallization of perovskites and their material properties were also investigated. Vasudevanpillai Biju et al. focused a laser beam at the precursor/gas interface, which

creates an optical potential to form local supersaturation and initiate the nucleation of perovskites.^{5,6} The grown perovskite single crystals exhibit good crystallinity and photoluminescence properties. However, the growth rate is low, i.e., it requires 1167 seconds to obtain a crystal of $200 \times 163 \mu\text{m}^2$ and the crystal shape cannot be accurately controlled. Optothermal effect was also harnessed to create local supersaturation for the precursor solution whose solubility is reduced at improved temperature.⁷ The fabricated perovskite polycrystalline wire has a fluorescence lifetime comparable to that of single crystals grown in bulk solution. However, it is not a general strategy as the solubility of some precursor solution is improved at higher temperature. Moreover, laser-triggered acid-catalyzed hydrolysis of N-methylformamide was employed to achieve local nucleation of perovskite.⁸ The obtained crystal exhibits a high photoconductive responsivity of $0.83 \text{ mA}\cdot\text{W}^{-1}$. Nevertheless, precise control of the crystallization pathway remains challenging to fabricate perovskite micropatterns.

According to the reviewer's suggestion, we have discussed the laser-controlled crystallization of perovskites in the introduction:

Main text, line 24, page 3 and lines 1-9, page 4:

“Laser-controlled crystallization can potentially address this fundamental problem. The key is to locally tune the supersaturation using a laser beam. For instance, crystallization of halide perovskites can be triggered in a laser-generated optical potential at the liquid/gas interface and the grown crystals exhibit good crystallinity and photoluminescence (PL) properties.^{32,33} The reduction of solubility in the laser controlled temperature field can also be harnessed for crystallization control, with the

fluorescence lifetime of the grown materials comparable to that of the single crystals grown in bulk solution.³⁴ Moreover, laser-triggered acid-catalyzed hydrolysis can also initialize the crystallization of halide perovskites.³⁵ Nevertheless, precise control of the crystallization pathway remains challenging to fabricate perovskite micropatterns.” *was added.*

Refs. 5-8 was added in the manuscript as Refs. 32-35, respectively.

2. Mechanistic details: the main text is rather vague about the mechanism, whereas the SI contains quite some insights. I would suggest moving part of this discussion to the main text.

Answer: Thank you very much for the reviewer’s suggestion. According to the reviewer’s suggestion, we have provided more discussion on the mechanism in the main text:

Main text, line 24, page 4 and lines 1-13, page 5:

“When a laser beam (515 nm femtosecond laser beam at 0.45 mW) is focused onto a perovskite crystal in the precursor solution, a thermal hot spot with a maximum temperature of 318 K is created around the perovskite crystal, with the maximum temperature tunable by the laser power (Figs. 1b and d-f). The local temperature gradient drives thermal diffusion, Marangoni convection, and interfacial evaporation in the precursor solution, which redistribute the molecular species around the laser spot (see Fig. 1g and detailed analysis in Supplementary Note 1).” *was replaced by* “When a laser beam (515 nm femtosecond laser beam at 0.45 mW, beam size 1.3 μm) is focused

onto a perovskite crystal in the precursor solution, the photon is absorbed by the crystals and part of the energy is converted to heat through non-radiative decay. Heat diffusion is limited by the low thermal conductivity of MAPbBr₃ (0.51 W/m·K) and DMF (0.1816 W/m·K).^{36,37} Consequently, a thermal hot spot with a maximum temperature of 318 K is created around the perovskite crystal, with the maximum temperature tunable by the laser power (Figs. 1b and d-f). As the thickness of precursor solution layer is thin, the local optical heating introduces temperature gradient at the liquid/gas interface and drives Marangoni convection (see Fig. 1g) and interfacial evaporation. It is noted that the local evaporation reduces the amount of solvent molecules and improve the solute concentration, which is the key factor for the formation of supersaturation environment. The evaporation rate increased with the local temperature due to higher saturated vapor pressure. The Marangoni convection, together with the non-isothermal diffusion, redistribute the molecular species in the solution and creates a local supersaturation environment around the laser spot (Supplementary Note 1).”.

Refs. 9, 10 *was added* as Refs. 36, 37 in the manuscript.^{9,10}

Main text, lines 3-17, page 7:

“At a low optical power (0.1 mW), a mild improvement of local precursor concentration accelerates the ionic reaction and leads to an improved spontaneous growth rate of the MAPbBr₃ crystal, with the solid-liquid interface under laser irradiation migrating at 0.33 μm/s to maintain the low surface energy (see Supplementary Video 1). Above 0.15 mW, a rapid liquid-to-solid phase transition leads to the exposure of atomically rough interfaces at the growth forefront with much higher surface energy (see detailed

analysis in Supplementary Note 2). Consequently, the growth rate is dramatically improved, which reaches 0.1 mm/s at 0.42 mW (see Supplementary Fig. 2).” *was replaced by* “At a low optical power (0.1 mW), a mild improvement of local supersaturation accelerates the ionic reaction and leads to an improved spontaneous growth rate of the MAPbBr₃ crystal. In this situation, the average time for bonding atoms in the precursor solution to the crystal surface t_b is significantly longer than the average time of atomic rearrangement process t_r . The rearrangement process occurs efficiently to form a flat atomic layer before formation of a second layer, leading to a layer-by-layer spontaneous growth mode. In the experiment, we observed the solid-liquid interface under laser irradiation migrating at 0.33 $\mu\text{m/s}$ to maintain the low surface energy (see Supplementary Video 1). Above 0.15 mW, the local concentration is higher and t_b is significantly reduced ($t_b < t_r$). There are many randomly bound solute atoms bonded to the crystal surface as the atoms cannot relax efficiently to maintain the low-energy surface. Such a rapid liquid-to-solid phase transition leads to the exposure of atomically rough interfaces at the growth interface with much higher surface energy, which reduces the thermodynamic barrier of liquid-to-solid phase transition and dramatically improves the growth rate (see detailed analysis in Supplementary Note 2). In our experiment, the growth rate reaches 0.1 mm/s at 0.42 mW (see Supplementary Fig. 2).”.

3. The claim on generality is overstated: The authors demonstrate the principle for three perovskite compounds, but the mechanism is likely not working for any compound. It

would help to be clear about the limitations, which can also be combined with the current insights on the mechanism.

Answer: Thank you very much for the reviewer's suggestion. As pointed out by the reviewer, it is not working for any compound. In principle, OCL is applicable to wet-chemistry synthesis of materials based on supersaturation-driven crystallization, if highly volatile solvent can be employed to dissolve the precursor solute and the grown materials own high optothermal conversion efficiency. If the optothermal conversion efficiency of the materials is low, high-power laser is required to create a high local temperature gradient, which may lead to irradiation damage on the crystal. Moreover, as the supersaturation is created by local evaporation of the solvent molecules, a highly volatile solvent is required for the preparation of precursor solution. However, the OCL approach cannot be applied to the wet-chemistry synthesis of materials based on other mechanism, e.g., co-precipitation synthesis, sol-gel method, etc.

According to the reviewer's suggestion, we have made revision in the main text:

Abstract, lines 13-15, page 2:

“The OCL technique is insensitive to material species and can be potentially extended to the fabrication of single-crystal structures of a variety of materials beyond halide perovskites, such as organic crystals or other chemically synthesized semiconductors, for diverse device applications.” *was replaced by* “The OCL technique can be potentially extended to the fabrication of single-crystal structures beyond halide perovskites, once crystallization can be triggered under the laser-directed local supersaturation.”.

Conclusion, lines 22-23, page 19 and lines 1-2, page 20:

“With its general applicability and diversity, the OCL approach can be potentially extended to the fabrication of single-crystal structures of a variety of materials beyond halide perovskites, such as organic crystals or other chemically synthesized semiconductors, for diverse device applications.” *was replaced by* “Considering the working principle, the OCL approach can be potentially extended to the fabrication of single-crystal structures of other materials based on supersaturation-driven crystallization, if highly volatile solvent can be employed to dissolve the precursor solute and the grown materials own high optothermal conversion efficiency.”.

4. Evidence for claim on surface quality: The authors frame the work around controlling defect density, morphology, and high surface quality. The control on growth at macroscopic scale is very clearly demonstrated, and impressive, but evidence for the quality on the surface quality etc. is rather minimal. Only one SEM in the SI shows macroscopic roughness, and Figure 3c shows only a zoomed-out SEM micrograph which makes it hard to judge the surface roughness. It would therefore be important to include a close-up SEM or AFM in the main text to show how smooth this surface is.

Answer: Thank you very much for the reviewer’s comments. We have provided close-up SEM images of the bull-shaped single crystal in Fig. R4 and Fig. 3c in the main text, which show high surface quality of the grown structures. In addition, we provided another high-magnification SEM image for a grown arrow pattern and checked the surface roughness using white light interferometry, which gives a value of 6.58 nm (Fig.

R5 and Supplementary Fig. 5).

Figure R4. SEM characterization of a bull-shaped single crystal. The SEM images in red dashed box and orange dashed box show the close-up images of the marked area in the left panel. Scale bar: 100 μm for left panel and 5 μm for the close-up images.

Figure R5. SEM and white light interferometry characterization of a grown MAPbBr₃ arrow pattern. **a**, Overall and high-magnification SEM images of the pattern. Scale bar: 10 μm . **b**, White light interferometry characterization of the pattern. The calculated roughness is $R_q = 6.58 \text{ nm}$. Scale bar: 10 μm .

We have added the close-up SEM images of the bull-shaped single crystal (Fig. R4) as Fig. 3c in the revised manuscript.

Supplementary Information, lines 60-65, page 5:

Fig. R5 *was added* in the Supplementary Materials as Supplementary Fig. 5.

Main text, lines 4-5, page 14 and lines 1-2, page 15:

“c, The SEM and EDS mapping of a bull-shaped single crystal. Scale bar: 100 μm .”
was replaced by “c, The SEM and EDS mapping of a bull-shaped single crystal. The SEM images in red dashed box and orange dashed box show the close-up images of the marked area in the top left panel. Scale bar: 100 μm for top left panel and 5 μm for the close-up images.”.

Main text, lines 10-13, page 15:

“From Fig. 3c, we can see that the crystal surface is clean and smooth, without observation of any grain boundary” *was replaced by* “From Fig. 3c, we can see that the crystal surface is clean and smooth, without observation of any grain boundary in close-up images. The surface roughness measured by white light interferometry gives a value of 6.58 nm (see Supplementary Fig. 5).”.

5. Experimental details: Some essential details are not clear/missing from the main text or SI. Please incorporate these details:

i) How is the temperature measured in/near the laser spot.

Answer: Thank you very much for the reviewer’s comments. The temperature distribution was recorded by Phasics SID4 camera integrated in a Nikon Ti2-U microscope. A 515 nm femtosecond laser of different optical powers was focused on the perovskite crystal. The phase change of the white light was recorded in-situ by Phasics SID4 camera to measure the refractive index change of the solvent, and the temperature variation is then derived via an inversion problem algorithm as the relation between the refractive index and the temperature is known. The working principle is

described in the literature “Thermal Imaging of Nanostructures by Quantitative Optical Phase Analysis”.¹¹

We have added the experimental details in the Methods section and added the reference as Ref. 46 in main text.

Methods, lines 5-10, page 24:

“The temperature distribution was recorded by Phasics SID 4” *was replaced by* “The temperature distribution was recorded by Phasics SID 4 camera integrated in a Nikon Ti2-U microscope. A 515 nm femtosecond laser of different optical powers was focused on the perovskite crystal. The phase change of the white light was recorded in-situ by Phasics SID4 camera to measure the refractive index change of the solvent, and the temperature variation is then derived via an inversion problem algorithm as the relation between the refractive index and the temperature is known.⁴⁶”

Ref.11 *was added as* Ref. 46 in the manuscript.

ii)How is concentration of the precursors selected. Is this done in simulations or experimentally? Is the temperature measured in the presence with the spot focused on a crystal and next to a crystal? Is the solution already initially supersaturated?

Answer: Thank you very much for the reviewer’s comments. The concentration of perovskite precursors is saturated to avoid spontaneous dissolution of the crystals after laser-directed printing. Due to the spontaneous evaporation, the solution will become supersaturated during the experiment. The saturation concentration of different halide perovskites can be obtained in Refs. 12-16 and it is done in experiment.¹²⁻¹⁶ The local

temperature was measured when the single crystal was irradiated by the focus laser spot.

We have added the experimental details in Methods:

Methods, lines 9-10, page 21:

“The saturation concentration of different halide perovskites is obtained from Ref. 26 and Refs. 42-45.” *was added.*

Methods, lines 5-10, page 24:

“The temperature distribution was recorded by Phasics SID 4” *was replaced by* “The temperature distribution was recorded by Phasics SID 4 camera integrated in a Nikon Ti2-U microscope. A 515 nm femtosecond laser of different optical powers was focused on the perovskite crystal. The phase change of the white light was recorded in-situ by Phasics SID4 camera to measure the refractive index change of the solvent, and the temperature variation is then derived via an inversion problem algorithm as the relation between the refractive index and the temperature is known.⁴⁶”

Methods, line 14, page 21: Ref. 12 was added in the revised manuscript as Ref. 26.

Methods, line 23, page 21: Ref. 13 was added in the revised manuscript as Ref. 42.

Methods, line 7, page 22: Ref. 14 was added in the revised manuscript as Ref. 43.

Methods, line 14, page 22: Ref. 15 was added in the revised manuscript as Ref. 44.

Methods, line 20, page 22: Ref. 16 was added in the revised manuscript as Ref. 45.

iii) One sentence is mentioned on the nucleation part, but this remains very unclear. It would be helpful to incorporate details on how you start the process, especially because the current mechanism can not explain exactly how the seed crystal forms.

Answer: Thank you very much for the reviewer's comments. The seed crystals can be obtained in two different ways to start OCL process.

(1) As the prepared precursor solution is saturated. The spontaneous evaporation can lead to spontaneous nucleation and provides some seed crystals.

(2) The nucleation can also be triggered by a focus laser spot at a high optical power of 15 mW, which creates a microbubble to concentrate the solute.¹⁷ Since the light absorption can be dramatically improved after nucleation. The irradiation time should be carefully controlled to avoid laser damage of the crystals.

We have added the experimental details in the Methods section.

Methods, lines 12-16, page 23:

“Due to spontaneous evaporation of the saturated precursor solution, spontaneous nucleation may occur and provides seed crystals to start the OCL process. In addition, seed crystals can also be created by irradiation of a high-power (e.g., 15 mW) femtosecond laser, which generated a microbubble to concentrate the precursor solute for nucleation. The irradiation time should be carefully controlled to avoid laser damage of the crystals.” *was added*.

iv) It would be helpful to provide details on the focal spot and power density in the main text. Also, the pulsed laser likely has much higher peak power, so a comparison between both illumination systems should be done carefully. For instance, does the crystal surface get damaged during the pulsed laser shots?

Answer: Thank you very much for the reviewer's comments. The spot diameter of the

focus laser beam is about 1.3 μm , and the typical optical intensity is $3.6 \times 10^4 \text{ W/cm}^2$. The optical intensity we use in the experiments is 1 orders of magnitude lower than that used in femtosecond laser ablation^{18,19}, which could avoid the irradiation damage on the crystals. Moreover, the reduction of repetition frequency (from 5 MHz to 0.3 MHz) with same optical intensity ($2.5 \times 10^4 \text{ W/cm}^2$) could significantly improve the surface quality and the photoluminescence properties, although the peak power is higher at low repetition frequency. As shown in Fig. R6, both the bright-field optical image and the confocal fluorescence image show that the structure fabricated at the repetition frequency 0.3 MHz is more uniform, while structures fabricated at 5 MHz shows a decrease of fluorescence intensity in the center. This suggests that the low repetition frequency is preferred to improve the quality of the grown perovskites. However, when the repetition frequency is lower (e.g., 200 kHz or 100 kHz), microbubble can be generated due to the higher peak power and laser-directed printing can be interrupted.

Figure R6. Optical (left) and confocal fluorescence images (right) of the structures fabricated by fs-laser with repetition frequencies of 0.3 MHz and 5 MHz, respectively. At the same optical intensity of 2.5×10^4 W/cm², femtosecond laser with high repetition frequency causes crystal damage, although the peak power is lower. Scale bar: 20 μ m.

We have added the beam size in main text to calculate the power density.

Main text, line 24, page 4 and line 1, page 5:

“When a laser beam (515 nm femtosecond laser beam at 0.45 mW) is focused onto a perovskite crystal in the precursor solution,” *was replaced by* “When a laser beam (515 nm femtosecond laser beam of 0.45 mW in optical power and 1.3 μ m in beam size) is focused onto a perovskite crystal in the precursor solution,”.

We have discussed the influence of repetition frequency in the main text.

Main text, lines 6-10, page 8:

“The repetition frequency of femtosecond laser also influences the quality of the grown structures. Normally, low repetition frequency is preferred in our experiments to suppress the defect generation (Supplementary Fig. 3). However, microbubble can be generated at lower repetition frequency (e.g., 0.1 MHz) due to the higher pulse energy and laser directed printing can be interrupted.” *was added.*

Supplementary Information, lines 46-50, page 4:

Fig. R6 *was added* in the Supplementary Materials as Supplementary Fig. 3.

v) The surfactant addition is very elegant. Do you also need to increase the laser power to induce nucleation and maintain growth?

Answer: Thank you very much for the reviewer’s comments. After the addition of ligands, the laser power should be slightly increased to drive the desorption of ligands on the surface and to maintain the stable crystal growth. For instance, the optical intensity to initialize the laser printing process is improved from 1.8×10^4 W/cm² to 2.4×10^4 W/cm² when the surfactant of 1:300 in concentration (the volume ratio between oleylamine and ligand-free precursor) is added. This can also be verified in the measurement of growth velocity at different optical powers, which can be seen in Fig. 2h in the manuscript (as Fig. R7). We can see that an optical power of ~ 400 μ W is required to achieve a growth velocity of 0.05 mm/s when surface ligands are absent. However, an optical power above 550 μ W is required to maintain the same velocity when the surfactant of 1:300 in concentration is added. The nucleation also requires a

higher optical power. For instance, laser-induced nucleation in the precursor without ligand is 10 mW (by using a $20\times$ objective, $NA = 0.5$, repetition frequency is 0.3 MHz), while it requires at least 12.5 mW when the ligands of 1:300 in concentration is added.

Figure R7. The maximum growth velocity of LDPSC as functions of the surface ligand concentration and the optical power.

We have added more discussion in the revised main text:

Main text, line 24, page 11 and lines 1-3, page 12:

“We can see that the single-crystal growth velocity is significantly decreased at an improved ligand concentration, which further verifies the passivation effect of the surface ligands” *was replaced by* “We can see that the single-crystal growth velocity is significantly decreased at an improved ligand concentration. From another perspective, the laser power has to be improved to maintain the crystal growth at the same velocity after the addition of surface ligands, which further verifies the passivation effect of the surface ligands.”

vi) Line 82 “The local solute concentration can be further tuned by the laser power”

This is not clear. Do you mean that the solution evaporates faster?

Answer: Thank you very much for the reviewer’s comments. When the laser power is increased, the local temperature is increased to accelerate the evaporation, the convective flow, and the non-isothermal diffusion. Thus, the local solute concentration (or supersaturation) can be significantly improved, which can also be revealed in the simulation (Supplementary Fig. 1, also see Fig. R8).

Figure R8. The simulation results of the supersaturation at different laser powers.

We have added more discussion in the supplementary information:

Supplementary Information, lines 198-199, page 14:

“When the laser power is increased, the local solute concentration (or supersaturation) is improved as it accelerates the evaporation, the convective flow, and the thermal diffusion.” *was added.*

vii)Line 119: “solubility is reduced at an improved temperature” I assume the authors mean an increased temperature. Please provide a reference or data that show that this is indeed the case.

Answer: Thank you very much for the reviewer’s suggestion. We have changed the terms and added the reference in the main text.

Main text, line 12, page 8:

“the solubility is reduced at an improved temperature” *was replaced by* “the solubility

is reduced at an increased temperature”.

Ref. 12 *was added* as Ref. 26 in the manuscript.¹²

Referee 3

Comments:

I co-reviewed this manuscript with one of the reviewers who provided the listed reports.

This is part of the Nature Communications initiative to facilitate training in peer review and to provide appropriate recognition for Early Career Researchers who co-review manuscripts.

Answer: Thank you very much for the reviewer's positive and insightful comments.

We have made point-to-point responses to the reviewer's comments as above.

References

- 1 Cui, X., Chen, G. & Han, X. Experimental vapor pressure data and a vapor pressure equation for N,N-dimethylformamide. *J. Chem. Eng. Data* **51**, 1860-1861 (2006).
- 2 Yoshikawa, H. Y. *et al.* Laser energy dependence on femtosecond laser-induced nucleation of protein. *Appl. Phys. A* **93**, 911-915 (2008).
- 3 Yuyama, K.-i. *et al.* A Single Spherical Assembly of Protein Amyloid Fibrils Formed by Laser Trapping. *Angew. Chem., Int. Ed.* **56**, 6739-6743 (2017).
- 4 Alexander, A. J. & Camp, P. J. Non-photochemical laser-induced nucleation. *J. Chem. Phys.* **150**, 040901 (2019).
- 5 Yuyama, K. i., Islam, M. J., Takahashi, K., Nakamura, T. & Biju, V. Crystallization of Methylammonium Lead Halide Perovskites by Optical Trapping. *Angew. Chem., Int. Ed.* **57**, 13424-13428 (2018).
- 6 Islam, M. J. *et al.* Mixed-halide perovskite synthesis by chemical reaction and crystal nucleation under an optical potential. *NPG Asia Mater.* **11**, 31 (2019).
- 7 Chou, S. S. *et al.* Laser Direct Write Synthesis of Lead Halide Perovskites. *J. Phys. Chem. Lett.* **7**, 3736-3741 (2016).
- 8 Arciniegas, M. P. *et al.* Laser-Induced Localized Growth of Methylammonium Lead Halide Perovskite Nano- and Microcrystals on Substrates. *Adv. Funct. Mater.* **27**, 1701613 (2017).
- 9 Haeger, T., Heiderhoff, R. & Riedl, T. Thermal properties of metal-halide perovskites. *J. Mater. Chem. C* **8**, 14289-14311 (2020).
- 10 Cai, G., Zong, H., Yu, Q. & Lin, R. Thermal conductivity of alcohols with acetonitrile and N,N-Dimethylformamide. *J. Chem. Eng. Data* **38**, 332-335 (1993).
- 11 Baffou, G. *et al.* Thermal Imaging of Nanostructures by Quantitative Optical Phase Analysis. *ACS Nano* **6**, 2452-2458 (2012).
- 12 Saidaminov, M. I., Abdelhady, A. L., Maculan, G. & Bakr, O. M. Retrograde solubility of formamidinium and methylammonium lead halide perovskites enabling rapid single crystal growth. *Chem. Commun. (Camb.)* **51**, 17658-17661 (2015).
- 13 Dang, Y. *et al.* Bulk crystal growth of hybrid perovskite material CH₃NH₃PbI₃. *CrystEngComm* **17**, 665-670 (2015).
- 14 Maculan, G. *et al.* CH₃NH₃PbCl₃ Single Crystals: Inverse Temperature Crystallization and Visible-Blind UV-Photodetector. *J. Phys. Chem. Lett.* **6**, 3781-3786 (2015).
- 15 Petrov, A. A., Ordinartsev, A. A., Fateev, S. A., Goodilin, E. A. & Tarasov, A. B. Solubility of Hybrid Halide Perovskites in DMF and DMSO. *Molecules* **26**, 7541 (2021).
- 16 Dirin, D. N., Cherniukh, I., Yakunin, S., Shynkarenko, Y. & Kovalenko, M. V. Solution-Grown CsPbBr₃ Perovskite Single Crystals for Photon Detection. *Chem. Mater.* **28**, 8470-8474 (2016).

- 17 Rajeeva, B. B. *et al.* Accumulation-driven unified spatiotemporal synthesis and structuring of immiscible metallic nanoalloys. *Matter* **1**, 1606-1617 (2019).
- 18 Tian, X., Wang, L., Li, W., Lin, Q. & Cao, Q. Whispering Gallery Mode Lasing from Perovskite Polygonal Microcavities via Femtosecond Laser Direct Writing. *ACS Appl. Mater. Interfaces* **13**, 16952-16958 (2021).
- 19 Tian, X. *et al.* Femtosecond laser direct writing of perovskite patterns with whispering gallery mode lasing. *J. Mater. Chem. C* **8**, 7314-7321 (2020).

REVIEWERS' COMMENTS

Reviewer #1 (Remarks to the Author):

The authors did a great job and addressed all comments raised by the reviewers. I would recommend publication of the manuscript in Nature Communications in its current form.

Reviewer #2 (Remarks to the Author):

The rebuttal and updated manuscript addressed all my concerns and I recommend publication. Three remarks that the authors may consider for finalising the manuscript:

1. Line 60. Without loss of generality, we experimentally demonstrate the laser-directed growth single-crystal halide perovskites into arbitrary micropatterns. I would omit the "Without loss of generality" as this is a very broad claim.

2. Fig.1 evaporation of the solution is integral to the process. Would it be helpful to add this to the schematic and the figure caption.

3. Line 149 solubility is increased with temperature

Reviewer #3 (Remarks to the Author):

I co-reviewed this manuscript with one of the reviewers who provided the listed reports as part of the Nature Communications initiative to facilitate training in peer review and appropriate recognition for co-reviewers.

Response to reviewer's comments

Reviewer #1 (Remarks to the Author):

Comments:

The authors did a great job and addressed all comments raised by the reviewers. I would recommend publication of the manuscript in Nature Communications in its current form.

Answer: Thank you very much for the reviewer's positive comments.

Reviewer #2 (Remarks to the Author):

The rebuttal and updated manuscript addressed all my concerns and I recommend publication. Three remarks that the authors may consider for finalising the manuscript:

Answer: Thank you very much for the reviewer's positive and valuable comments. We have made point-to-point response to the comments:

1. Line 60. Without loss of generality, we experimentally demonstrate the laser-directed growth single-crystal halide perovskites into arbitrary micropatterns. I would omit the "Without loss of generality" as this is a very broad claim.

Answer: Thank you very much for the reviewer's suggestion. According to the reviewer's suggestion, we have deleted the terms "Without loss of generality":

Main text, lines 73-74, page 4:

"Without loss of generality, we experimentally demonstrate the laser-directed growth single-crystal halide perovskites into arbitrary micropatterns." *was replaced by* "We experimentally demonstrate the laser-directed growth single-crystal halide perovskites into arbitrary micropatterns."

2. Fig.1 evaporation of the solution is integral to the process. Would it be helpful to add this to the schematic and the figure caption.

Answer: Thank you very much for the reviewer's suggestion. According to the reviewer's suggestion, we have added the interfacial evaporation in the schematic

(Fig.1a) and revised the figure caption.

Main text, lines 103-105, page 6:

“Schematic of OCL. The perovskite structure growth along the track of laser spot.” *was replaced by* “Schematic of OCL. The perovskite structure grows along the track of laser spot, where supersaturation is created by the coordinated effect of interfacial evaporation and convection.”.

3. Line 149 solubility is increased with temperature

Answer: We have revised the sentence as suggested by the reviewer.

Main text, line 153, page 8:

“whose solubility is improved with temperature” *was replaced by* “whose solubility is increased with temperature”.

Reviewer #3 (Remarks to the Author):

I co-reviewed this manuscript with one of the reviewers who provided the listed reports as part of the Nature Communications initiative to facilitate training in peer review and appropriate recognition for co-reviewers.

Answer: Thank you very much for the reviewer's valuable comments. We have made point-to-point response to the reviewer's comments as above.